# Sex Differences in Survival from Neuroendocrine Neoplasia in England 2012–2018: A Retrospective, Population-Based Study

**DOI:** 10.3390/cancers15061863

**Published:** 2023-03-20

**Authors:** Benjamin E. White, Beth Russell, Sebastiaan Remmers, Brian Rous, Kandiah Chandrakumaran, Kwok F. Wong, Mieke Van Hemelrijck, Rajaventhan Srirajaskanthan, John K. Ramage

**Affiliations:** 1Basingstoke and North Hampshire Hospital, Hampshire Hospitals NHS Foundation Trust, Basingstoke RG24 9NA, UK; 2Translational Oncology and Urology Research, School of Cancer & Pharmaceutical Sciences, King’s College London, London WC2R 2LS, UK; 3Department of Urology, Erasmus MC Cancer Institute, University Medical Center Rotterdam, 3015 GD Rotterdam, The Netherlands; 4NHS Digital, 7 and 8 Wellington Place, Leeds LS1 4AP, UK; 5King’s Health Partners ENETS Centre of Excellence, King’s College Hospital, London SE5 9RS, UK

**Keywords:** neuroendocrine tumour, neuroendocrine tumour, neuroendocrine neoplasia, carcinoid, epidemiology, survival, incidence, predictors of survival

## Abstract

**Simple Summary:**

We conducted a retrospective, population-based study comparing overall survival (OS) between males and females with neuroendocrine neoplasia (NEN). In total, 14,834 cases of NEN recorded in England’s National Cancer Registry and Analysis Service (NCRAS)), were analysed. Multivariable analysis, restricted mean survival time and mediation analysis were performed. Females displayed increased survival irrespective of the stage, morphology or level of deprivation, which was statistically significant in NEN of the lung, pancreas, rectum and stomach (*p* < 0.001). Stage of tumour mediated improved survival in stomach, lung, and pancreatic NEN but not in rectal NEN. Females diagnosed with NEN tend to survive longer than males, and stage at presentation only accounts for part of this effect. Future research in NEN, as well as prognostication and treatment, should consider sex as an important factor.

**Abstract:**

Pre-clinical studies have suggested sex hormone signalling pathways may influence tumorigenesis in neuroendocrine neoplasia (NEN). We conducted a retrospective, population-based study to compare overall survival (OS) between males and females with NEN. A total of 14,834 cases of NEN diagnosed between 2012 and 2018, recorded in England’s National Cancer Registry and Analysis Service (NCRAS), were analysed. The primary outcome was OS with 5 years maximum follow-up. Multivariable analysis, restricted mean survival time and mediation analysis were performed. Appendiceal, pulmonary and early-stage NEN were most commonly diagnosed in females; stomach, pancreatic, small intestinal, colonic, rectal and later-stage NEN were more often diagnosed in males. Females displayed increased survival irrespective of the stage, morphology or level of deprivation. On average, they survived 3.62 (95% CI 1.73–5.90) to 10.26 (6.6–14.45) months longer than males; this was statistically significant in NEN of the lung, pancreas, rectum and stomach (*p* < 0.001). The stage mediated improved survival in stomach, lung, and pancreatic NEN but not in rectal NEN. The reasons underlying these differences are not yet understood. Overall, females diagnosed with NEN tend to survive longer than males, and the stage at presentation only partially explains this. Future research, as well as prognostication and treatment, should consider sex as an important factor.

## 1. Introduction

Neuroendocrine neoplasia (NEN) are tumours arising from neuroendocrine cells; they share the traits of both nervous and endocrine cells and can release hormones in response to neuronal stimuli. NEN can be classified as well-differentiated neuroendocrine tumours (NET) or poorly differentiated neuroendocrine carcinoma (NEC), which include large and small cell differentiation. Although NEN can occur anywhere in the body, the majority arise in the gastrointestinal tract and lungs [1,2]. The symptomatology can be highly variable and is dependent on the tumour burden and hormone-secreting capacity [3].

The incidence of NEN is increasing globally [4]; theories to explain this include increased clinical awareness, more widespread availability of imaging techniques and endoscopy, and a possible ‘real’ increase. Risk factors for developing NEN include a family history of cancer, type 2 diabetes, obesity, cigarette smoking and alcohol intake. These findings are from retrospective case–control studies; high-quality prospective trials to identify risk factors have not yet been performed in NEN [5,6,7]. 

Survival at each NEN tumour site has improved in England over the last 25 years. This is likely due to a combination of increased detection of low-stage tumours resulting in ‘stage shift’ and the effect of treatment advances [8]. The predictors of survival at diagnosis of NEN identified so far are age, sex, organ site, stage, grade, deprivation (also known as socio-economic status) and marital status. As in other solid organ cancers, there are survival differences by sex in NEN. Population-based studies from North America that include large numbers of tumours have shown males to have statistically significant worse overall survival (OS), with HRs up to 1.26–1.27 for males compared to females [9,10]. Large cohort analyses from other population-based databases in England, Canada, Australia, Norway, Taiwan and others also demonstrated statistically significant worse OS in males in multivariable analysis [11,12,13,14,15].

Overall, sex plays an important role in survival from solid organ cancers [16]. Sex is a key modifier of pathophysiology via genetic, epigenetic and hormonal regulation. A different biological sex environment is created by genetic heterogeneity at the molecular level. Evidence shows that sex hormones affect cellular responses by modifying DNA expression, which in turn leads to different cell surface receptor expression [17]. Not only does this result in differing predispositions to and the manifestation of malignancy but it also affects the response to cancer therapy. Societal factors also play a role by influencing behaviours such as diet, smoking and physical activity, which in turn can influence health outcomes. Perceived gender also affects how a patient is treated both by society and clinicians [16]. 

Male predominance in solid organ cancers that affect both sexes has been observed worldwide [18]. Males are known to have greater exposure to risk factors, such as occupational, alcohol intake and smoking risk factors [19]. Survival is shorter for males across multiple solid organ cancer types [20]. As described above, these differences may be explained by sex-specific biology having effects on tumorigenesis, the stimulatory effects of androgens in male individuals and the protective effects of oestrogens in females seen in non-reproductive cancers [21], in addition to the influence of the societal, cultural or behavioural effect of gender roles.

Several pre-clinical studies have shown that the expression of oestrogen and progesterone receptors is associated with favourable outcomes amongst patients with gastroenteropancreatic NEN (GEP-NEN). Pancreatic NEN (pNEN) with higher oestrogen receptor-β expression are associated with a more favourable prognosis [22]. Females, especially those who are pre-menopausal, have a lower risk of mesenteric metastasis in small intestinal NEN (SI-NEN). SI-NEN have increased oestrogen and androgen receptor expression compared to normal tissue, suggesting that sex hormone signalling pathways may modulate metastatic potential [23,24,25]. Immunohistochemical assessment of progesterone receptor (PR) status may help to identify GEP-NEN with the potential for more aggressive behaviour [26,27]. Histological differences between males and females in lung carcinoids have also been identified [28]. A clinical trial is investigating the effect of tamoxifen in well-differentiated NEN based on oestrogen and progesterone receptors being expressed in around 20% of NEN [29].

We aimed to use restricted mean survival time (RMST) and mediation analysis to compare survival by sex in NEN and examine the influence of the stage on survival outcomes. To our knowledge, there are no other studies that have yet analysed sex differences in NEN in this way.

## 2. Methods

### 2.1. Data Source

This work utilised data from the National Cancer Registry and Analysis Service (NCRAS) of England, which captures over 99% of tumours recorded in England’s National Health Service [30,31]. The data were collected for individuals aged 16 and above who had been diagnosed with NEN between 2012 and 2018. The NCRAS database is updated as histopathological classification systems change, which presents a challenge in a rapidly evolving field such as NEN. Stage is recorded by NCRAS according to the European Neuroendocrine Tumor Society (ENETS) system for foregut [32] and mid- and hindgut [33] tumours and uses the Union for International Cancer Control tumour, node and metastasis system (UICC TNM) [34] for other sites. 

### 2.2. NEN Classification and Analytic Process

NEN occurring at all anatomical sites between C00 and C80 and malignant neoplasms of all sites (excluding haematological malignancy), according to the 10th edition of the WHO International Classification of Disease (ICD-10) were included. The morphology codes included 8013 (excluding lung [C34 and C78]), 8041–8045 (excluding lung), 8150–8158, 8240–8247, 8249 and 9091, according to the WHO International Classification of Diseases for Oncology, 3rd Edition (ICD-O-3) [35] in line with previously published work on NEN based on NCRAS data [13]. Large cell neuroendocrine and small cell carcinomas of the lung were excluded to enable a comparison with previous analyses and because the high incidence in this organ due to smoking would skew the results. Goblet cell adenocarcinomas (GCA) (ICD-O-3: 8243) were excluded from the dataset in view of their reclassification as non-NEN [36]. Duplicate tumours and tumours recorded as ‘death certificate only’, which made up less than 0.1% of the tumours, were excluded [37,38].

Mixed neuroendocrine non-neuroendocrine neoplasms (MiNEN) (formerly termed mixed adenoneuroendocrine carcinomas (MANEC)) and Merkel cell tumours were excluded. Only tumours diagnosed from 2012 onwards were included in the main survival analysis due to markedly improved coding and classification in recent years; unclassified stage tumours (25.6%) were excluded. It was decided that imputing missing data was not desirable due to a risk of bias in the resulting dataset [39].

Site groups were created from histological codes. The main sites were defined as the appendix, caecum, colon, lung, pancreas, rectum, small intestine or stomach, in line with other series [40]. Tumours with a primary site not registered as one of these ‘main’ primaries were excluded in order to clearly define the cohort and avoid inaccuracy in analysing the likely metastatic sites.

We, therefore, grouped the NEN morphologically, either as well-differentiated neuroendocrine tumours (NET) or poorly differentiated neuroendocrine carcinomas (NEC), similar to other recently published work [41]. The tumours classified as NET included carcinoids of typical, atypical, tubular and other well-differentiated neoplasms such as insulinoma and glucagonoma. The NEC included all the carcinomas and tumours with large and small cell neuroendocrine differentiation. Although all tumours have a histopathological classification, the Ki-67 index was not yet available on the NCRAS database at the time of the data transfer. 

The available variables suitable to be included in the analysis were site, age, sex, index of multiple deprivation (IMD), morphology and stage. The IMD is a measure of relative deprivation for small areas of England (lower layer super output areas, LSOA) and is composed of seven domains with relative weights: income (22.5%), employment (22.5), education (13.5%), health (13.5%), crime (9.3%), housing (9.3%) and environment (9.3%). 

### 2.3. Statistical Analytic Approach

The categorical variables were presented as percentages; the continuous variables were reported as the median and interquartile range (IQR). The primary endpoint was OS, calculated from the date of diagnosis and censored on 31 March 2020 and calculated using the Kaplan-Meier estimator with a maximum of 5 years follow-up. The 95% confidence interval (95% CI) was specified for all the results. All the variables were included in the multivariable analysis except for ethnicity. Ethnicity was excluded due to skewed data.

Cox regression multivariable analysis included sex, morphology, age group, stage, site and deprivation. Of these, sex and deprivation met proportional hazards assumptions. The other variables did not strictly meet proportional hazards assumptions and were therefore included in the final multivariable model as covariates with a time-varying effect (TVC). There was no multicollinearity between the variables. The accelerated failure time (AFT) models were tested for significance against the null models (Cox) using a likelihood ratio test (*p* < 0.001). We aimed to use age-adjusted restricted mean survival time (RMST) as a method to compare survival between the sexes. RMST is defined as the area under the survival curve up to a specific time point and can overcome some of the limitations of proportional hazard modelling [42]. 

The stage at presentation might explain some of the survival differences observed in NEN between males and females. Early-stage appendiceal and lung NEN, for example, occur more frequently in females, whilst late-stage pancreatic and stomach NEN occur more frequently in males [8]. Mediation analysis can be used to further study the relationship between sex and survival and how this is influenced by the stage [43]. Mediation analysis looks at how the relationship between an exposure (e.g., sex) and an outcome (e.g., survival) might be mediated by another variable (e.g., stage) whilst adjusting for other confounding factors (e.g., morphology, deprivation). 

RMST and age-adjusted RMST were calculated using the *strmst2* command in STATA. Mediation analysis was performed using the *med4way* command in STATA. For the mediation analysis, as mediators in med4way can be either continuous or dichotomous, stages I and II were classed as ‘early’ stage and stages III and IV were classed as ‘late’ stage. The statistical analyses and plots were performed using STATA/MP 16.0 (College Station, TX, USA: StataCorp LLC) and R.

## 3. Results

In total, 14,834 tumours recorded on the NCRAS database between 2012 and 2018 were eligible for analysis. The largest proportion of tumours occurred in the 65–74 age group, with a median age for the cohort of 65 (IQR 53-73) (Table 1). Closely matching the ethnic mix of England, the most frequent ethnicity was White (89%), followed by Asian (2.9%) and Black (2.3%).

The most common primary site was the lung (4661; 31.4% of tumours), followed by the small intestine (3201; 21.6%), pancreas (2183; 14.7%) and appendix (2146; 14.5%). Most tumours in the cohort were either stage I (5040; 34.0% of tumours) or stage IV (5121; 34.5%), with stages II and III being less frequent. There were 11,080 NET (74.7%) and 3754 NEC (25.3%). The tumours were spread relatively evenly across deprivation quintiles (20.5% to 18.3%) (Table 1).

Overall, there were slightly more females than males diagnosed with NEN (51.5% female vs. 48.5% male) (Table 2). The median age was similar in males and females (65.5 vs. 65). The youngest age group displayed female predominance (60.9%), but this disparity ceased above age 54 where the tumours became more evenly distributed. 

The appendix and lung sites were predominantly diagnosed in females (61.3% and 60.2% female), but stomach, pancreas, small intestine, colon and rectal NEN were most frequently diagnosed in males (56.3–61.9%). Stage I and II tumours showed a female preponderance (60.4% and 53.6% female). However, there were more stage III and IV tumours diagnosed in males (53.4% and 55.6% male). The sex distribution was relatively equal across all the deprivation quintiles. There were more NET diagnosed in females (54.0% female). However, the opposite was true in NEC, where 56.0% of the diagnoses occurred in males.

As expected, increasing age was associated with progressively increased hazard ratios (HR); compared to the <30 age group, the HR was 4.41 (95% CI 2.88–6.74) for the 30–54 age group, 5.35 (3.50–8.18) for the 55–65 age group, 6.13 (4.01–9.37) for the 65–74 age group and 7.72 (5.06–11.80) for those over 75. The rectum 1.27 (1.15–1.41) and stomach 1.26 (1.14–1.33) had the highest HRs of any of the main sites when compared to the appendix. Increasing stage was associated with increasing hazard, and the same pattern was observed with increasing deprivation quintile. A NEC was associated with a significantly increased HR compared to a NET (HR 1.29 (1.25–1.33)) (Table 3).

The age-adjusted 5-year RMST (Table 3) of the main sites showed that females displayed a survival advantage ranging from 3.62 (95% CI 1.73 to 5.90) to 10.26 (6.6 to 14.45) months. The exception was colonic NEN, which showed a male survival advantage, but this was not statistically significant. The sites where females showed a statistically significant improved survival were the lung, pancreas, rectum and stomach (all *p* < 0.001). Females had an increased survival at all tumour stages: 1.2 months (0.48 to 1.92) for stage I, 3.24 months (1.76 to 4.72) for stage II, 2.23 months (0.58 to 3.89) for stage III and 3.19 months (1.84 to 4.54) for stage IV. All these results were statistically significant (*p* < 0.001 except Stage III *p* = 0.008). Compared to the males, females survived longer when diagnosed with both morphological groups of NET or NEC, with an HR of 2.44 (1.74 to 3.13) and 4.92 (3.46 to 6.37) months, respectively (*p* < 0.001). Similarly, females had a longer survival in all the deprivation quintiles (4.45 to 5.14 months, *p* < 0.001).

Four-way decomposition mediation analysis of the four main primary sites found females to have a statistically significant survival advantage. Table 4 shows that females are less likely to be diagnosed at a later stage than males. Consistent with the age-adjusted RMST findings, females survived longer than males in all four sites according to the model for the outcome at each site. The stage mediated improved survival in females to a significant extent (37%) in stomach NEN and moderately (20%) in lung and pancreas NEN. The stage did not play a role (4%) in mediating survival in rectal NEN (Figure 1). 

## 4. Discussion

This study demonstrates that females diagnosed with neuroendocrine neoplasia display a survival advantage compared to males. Sex is a statistically significant predictor of survival in multivariable analysis [8]. The survival advantage for females remained statistically significant when examining the subgroups (Table 3). When examining the primary sites, those sites with a statistically significant increase in overall survival in females are the lung, pancreas, rectum and stomach. 

Mediation analysis suggested that the stage is responsible for the survival advantage seen in females to different extents depending on the primary site. The stage (early or late) at diagnosis appears to explain the survival advantage in stomach NEN to a large extent, in lung and pancreas NEN moderately and not at all in rectal NEN. The reasons underlying these differences are not yet understood. The stage is, therefore, an important intermediate (i.e., a mediator) on the pathway of the association between sex and survival and might help direct investigation of underlying causes. 

Although we have demonstrated that the stage partially explains differential survival according to sex in NEN, it is not clear why females tend to be diagnosed earlier than males with lung NEN, whilst the reverse is true for pancreatic and stomach NEN. Females were also observed to survive longer than males, even when diagnosed with the same stage and morphological type of NEN. Suggested explanations put forward for this in the past include biological reasons (including genetic, hormonal and other factors) [17,18] and environmental reasons (including behavioural, societal and cultural factors) [16]. Other country or health system-specific factors may also play a role, such as screening programmes for cancer or prominent public health campaigns. 

As described previously, there is an increasing body of pre-clinical research seeking to explain the sex difference in NEN. The expression of oestrogen and progesterone receptors, or sex hormone signalling pathways, may play a role in the differing biology between the sexes and tumorigenesis [22,23,24,25,26,27].

We suggest that future research could try to explain why females are presenting with earlier-stage NEN of the lung and males are presenting with later-stage NEN of the pancreas and stomach. This could be examined by looking at the mode of presentation or diagnosis, not sufficiently complete for us to analyse. The linkage of general practice records at the patient level, which would enable a richer analysis of risk factors such as smoking and obesity, might also help to explain the difference [44].

At presentation, it would be difficult to distinguish late-stage NEN, which have been slow-growing and undiscovered but may have transformed recently from aggressive ones, which have only been present for a short time. NEN could have had differing durations of exposure to an oncogenic microenvironment, which might mean a greater impact of sex differences, for example, the duration of exposure to sex hormones. To investigate this, a study model could be devised to predict the risk of the development of NEN compared to a background rate in the population before and after menopause. Since NEN are thought to be slow-growing tumours, commonly with the tumour being present both before and after menopause, this study design may be complex [25].

The use of Ki-67 in future studies would be beneficial, allowing for the mitotic index to be taken into account when analysing the differences in survival. Ki-67 has been recorded in the NCRAS database from 2020 onwards, reflecting how histological classification has developed over time, representing a good opportunity for further investigation [45]. Another model to further characterise how sex influences survival in NEN might be to retrospectively examine the differences between patients with a co-diagnosis of NEN and prostate cancer, having or not having antiandrogen therapy, as the role of androgen deprivation in survival from these tumours is unclear [46].

The limitations of this retrospective, population-based study include a historic lack of quality recorded data before 2012, particularly regarding the stage and morphology, meaning it was not possible to accurately compare these findings to earlier time periods. We had to rely on morphology to characterise the tumours, without a grade or Ki-67, again due to the incompleteness of the data. It was not possible to accurately analyse the diagnostic imaging pathways, chemotherapy treatments or health system routes to diagnosis due to the incompleteness of the data. However, this is improving over time as the NCRAS database becomes more complete.

## 5. Conclusions

We have demonstrated that females diagnosed with NEN tend to survive longer than males. The stage at presentation is only partially responsible for this difference and does not explain the underlying causes. It is not possible in this analysis to demonstrate causality with respect to sex hormones or other sex differences, such as treatment histories, which may be influencing this relationship. More research is needed to understand how sex affects presentation, disease progression and treatment response in NEN. Our research suggests that prognostication and treatment should take sex and gender into account. We suggest that future trials in NEN should consider and report on sex and gender throughout the research process.

## Figures and Tables

**Figure 1 cancers-15-01863-f001:**
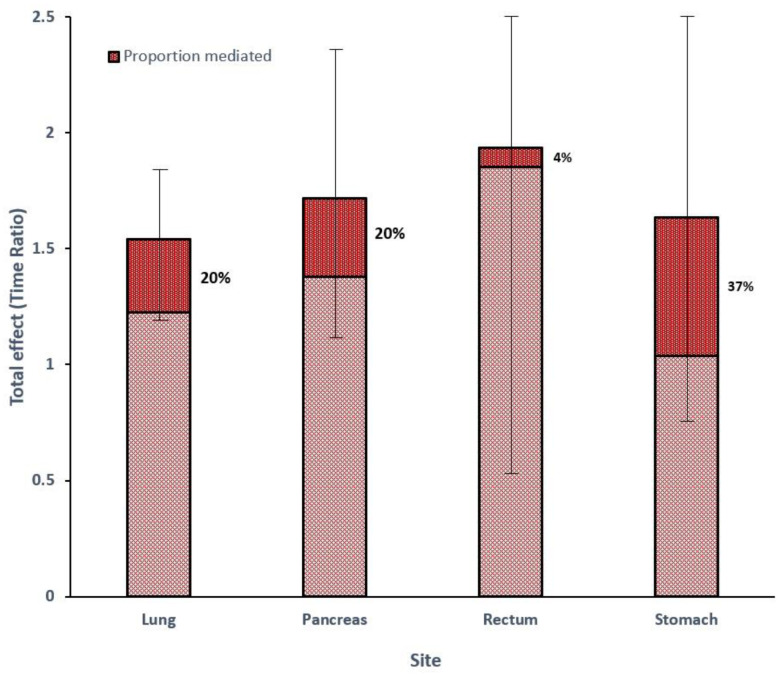
Total effect (Time ratio) by site in four-way decomposition mediation analysis. Proportion of the total effect mediated by stage shown as dark red area within bar. Error bars shown for total effect.

**Table 1 cancers-15-01863-t001:** Demographics of 14,834 NEN occurring at main organ primary sites 2012–2018. IQR, interquartile range; NEC, neuroendocrine carcinoma; NET, neuroendocrine tumour. Sex differential of NEN by main organ primary, stage and morphology see Appendix A.

Total		14,834	
Age (Median, IQR)		65 (53–73)	
		*n*	%
Age group	0–30	909	6.1%
	31–54	3141	21.2%
	55–64	3117	21.0%
	65–74	4450	30.0%
	75+	3217	21.7%
Ethnicity	Asian	425	2.9%
	Black	338	2.3%
	Mixed race	64	0.4%
	Other	192	1.3%
	White	13,197	89.0%
	Not stated	618	4.2%
Site	Appendix	2146	14.5%
	Caecum	528	3.6%
	Colon	509	3.4%
	Lung	4661	31.4%
	Pancreas	2183	14.7%
	Rectum	948	6.4%
	Small intestine	3201	21.6%
	Stomach	658	4.4%
Stage	I	5040	34.0%
	II	2004	13.5%
	III	2669	18.0%
	IV	5121	34.5%
Morphology	NET	11,080	74.7%
	NEC	3754	25.3%
Deprivation quintile	1—least deprived	3048	20.5%
	2	3194	21.5%
	3	3148	21.2%
	4	2722	18.3%
	5—most deprived	2722	18.3%

**Table 2 cancers-15-01863-t002:** Demographics of 14,834 NEN by sex differential. IQR, interquartile range; NET, neuroendocrine tumour; NEC, neuroendocrine carcinoma.

			Age Group					Ethnicity					
	Total 14,834	Median Age (IQR)	0–30	31–54	55–64	65–74	75+	Asian	Black	Mixed Race	Other	White	Not Stated
Male	7196	65.5 (54–73)	355	1447	1562	2257	1575	218	145	30	101	6389	313
	48.5%		39.1%	46.1%	50.1%	50.7%	49.0%	51.3%	42.9%	46.9%	52.6%	48.4%	50.6%
Female	7638	65 (51–73)	554	1694	1555	2193	1642	207	193	34	91	6808	305
	51.5%		60.9%	53.9%	49.9%	49.3%	51.0%	48.7%	57.1%	53.1%	47.4%	51.6%	49.4%
		**Site**								**Stage**			
		**Appendix**	**Caecum**	**Colon**	**Lung**	**Pancreas**	**Rectum**	**Small Intestine**	**Stomach**	**I**	**II**	**III**	**IV**
Male		831	228	298	1854	1230	544	1804	407	1996	929	1425	2846
		38.7%	43.2%	58.5%	39.8%	56.3%	57.4%	56.4%	61.9%	39.6%	46.4%	53.4%	55.6%
Female		1315	300	211	2807	953	404	1397	251	3044	1075	1244	2275
		61.3%	56.8%	41.5%	60.2%	43.7%	42.6%	43.6%	38.1%	60.4%	53.6%	46.6%	44.4%
		**Deprivation Quintile**					**Morphology**						
		**1—least deprived**	**2**	**3**	**4**	**5—most deprived**	**NET**	**NEC**					

Male		1524	1547	1488	1354	1283	5093	2103					
		50.0%	48.4%	47.3%	49.7%	47.1%	46.0%	56.0%					
Female		1524	1647	1660	1368	1439	5987	1651					
		50.0%	51.6%	52.7%	50.3%	52.9%	54.0%	44.0%					

**Table 3 cancers-15-01863-t003:** Multivariable analysis of survival in the cohort with hazard ratios (HR), and analysis of sex differences in 5-year restricted mean survival time (RMST), stratified by variable. Nb. age not displayed in table due to use of age-adjusted RMST. However, results of age in multivariable analysis are included in text.

Multivariable Survival Analysis			Analysis of Sex Difference in Survival		
			5-year RMST		Age-Adjusted Female Survival Advantage
Variable		HR (95%CI)	Sex	Months (95% CI)	Months (95% CI), *p*-Value
Site	Appendix	Reference	M	4.64 (4.56–4.72)	0.95 (−0.18 to 2.10), *p* = 0.098
*p* = 0.041			F	4.76 (4.71–4.82)	
	Caecum	1.01 (0.9–1.11)	M	3.11 (2.83–3.39)	2.77 (−1.61 to 7.15), *p* = 0.215
			F	3.30 (3.07–3.54)	
	Colon	1.14 (1.04–1.25)	M	2.22 (1.98–2.47)	−2.44(−6.90 to 2.04), *p* = 0.287
			F	1.99 (1.71–2.28)	
	Lung	1.22 (1.13–1.31)	M	2.96 (2.86–3.06)	9.85 (8.40 to 11.30), *p* < 0.001
			F	3.76 (3.69–3.83)	
	Pancreas	1.18 (1.09–1.28)	M	3.16 (3.05–3.28)	3.62 (1.73 to 5.90), *p* < 0.001
			F	3.52 (3.39–3.65)	
	Rectum	1.27 (1.15–1.41)	M	3.24 (3.05–3.42)	5.68 (2.38 to 8.96), *p* < 0.001
			F	3.75 (3.55–3.94)	
	Small intestine	0.94 (0.87–1.02)	M	4.01 (3.93–4.09)	1.31 (−0.16 to 2.75), *p* = 0.081
			F	4.09 (4.00–4.18)	
	Stomach	1.26 (1.14–1.33)	M	2.18 (1.97–2.38)	10.26 (6.6 to 14.45), *p* < 0.001
			F	3.22 (2.95–3.49)	
Stage	I	Reference	M	4.66 (4.62–4.71)	1.2 (0.48 to 1.92), *p* < 0.001
*p* < 0.001			F	4.74 (4.71–4.78)	
	II	1.38 (1.29–1.46)	M	4.60 (4.53–4.67)	3.24 (1.76 to 4.72), *p* < 0.001
			F	4.25 (0.05–4.35)	
	III	1.58 (1.49–1.68)	M	3.86 (3.77–3.95)	2.23 (0.58 to 3.89), *p* = 0.008
			F	4.07 (3.97–4.16)	
	IV	2.11 (2.01–2.23)	M	2.02 (1.94–2.10)	3.19 (1.84 to 4.54), *p* < 0.001
			F	2.28 (2.19–2.37)	
Morphology	NET	Reference	M	4.20 (4.15– 4.24)	2.44 (1.74 to 3.13), *p* < 0.001
*p* < 0.001			F	4.43 (4.39–4.46)	
	NEC	1.29 (1.25–1.33)	M	1.52 (1.45–1.60)	4.92 (3.46 to 6.37), *p* < 0.001
			F	1.94 (1.85–2.04)	
Deprivation	1—least deprived	Reference	M	3.53 (3.43–3.64)	5.14 (3.49 to 6.78), *p* < 0.001
*p* < 0.001			F	4.02 (3.93–4.11)	
	2	1.11 (1.02–1.21)	M	3.36 (3.26–3.46)	5.68 (4.00 to 7.36), *p* < 0.001
			F	3.90 (3.81–3.99)	
	3	1.09 (1.02–1.19)	M	3.46 (3.35–3.56)	5.16 (3.48 to 6.84), *p* < 0.001
			F	3.91 (3.82–4.00)	
	4	1.21 (1.11–1.33)	M	3.34 (3.23–3.46)	4.95 (3.32 to 9.98), *p* < 0.001
			F	3.87 (3.77–3.97)	
	5—most deprived	1.32 (1.22–1.45)	M	3.32 (3.20–3.43)	4.45 (2.58 to 6.31), *p* < 0.001
			F	3.68 (3.58–3.78)	

**Table 4 cancers-15-01863-t004:** Mediation analysis of the four significant sites of NEN using exposure as sex, outcome as survival and mediator as stage group (early or late). *n*, number of tumours;%F, proportion of females; TR, time ratio; OR, odds ratio; CDE, controlled direct effect; INT, interaction; IE, indirect effect; MED, mediation.

					Four-Way Decomposition				
Site	*n*	%F	Model for Outcome (TR) (95% CI)	Model for Mediator (OR) (95% CI)	Total Effect (TR) (95% CI)	CDE%	INT%	IE%	MED%
Lung	4661	60	1.43 (1.16–1.76)	0.61 (0.53–0.71)	1.54 (1.22–1.86)	97%	−12%	14%	20%
Pancreas	2183	44	1.70 (1.16–2.50)	0.81 (0.67–0.97)	1.72 (1.13–2.30)	170%	−80%	10%	20%
Rectum	948	43	1.91 (0.93–3.91)	0.78 (0.52–1.19)	1.94 (0.55–3.32)	107%	−9%	2%	4%
Stomach	658	38	1.50 (0.88–2.56)	0.48 (0.32–0.73)	1.64 (0.82–2.45)	103%	−24%	20%	37%

## Data Availability

The datasets presented in this article are not readily available as NCRAS data is restricted by law.

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
