# Peer review of "Sex Differences in Survival from Neuroendocrine Neoplasia in England 2012–2018: A Retrospective, Population-Based Study"

_cancers, 2023, doi:10.3390/cancers15061863_

Round 1
Reviewer 1 Report
This article entitled with " Sex differences in survival from neuroendocrine neoplasia in England 2012-2018: a retrospective, population-based study” showed that females diagnosed with neuroendocrine neoplasms tend to survive longer than males by using restricted mean survival time and mediation analysis.
There are no specific comments.
Author Response
Thank you for your review.
Reviewer 2 Report
The study by White et al. is a retrospective study. The authors used the information of patients diagnosed with neuroendocrine neoplasia (NEN) between 2012 and 2018 in the database from the National Cancer Registry and Analysis Service (NCRAS) to analyze the gender difference in the overall survival of patients with NEN. The authors concluded that females diagnosed with NEN tend to survive longer than males. However, they also pointed out that their conclusion doesn’t explain underlying causes of gender difference observed.
The gender differences in cancer susceptibility and cancer survival for many types of cancers have been reported since 2000. This manuscript is well-written in reviewing NEN and describing the data source and analysis approaches and results. However, there is not much in-depth progress in studying the role of gender difference in cancer survival. In addition, it is not clear whether the gender differences in NEN patients concluded in this study can be sole explained by function of sex hormones as proposed by the authors.
Overall, I feel that this study is not completed yet at this stage.
Specific comments:
1. The study described in the manuscript lacks in-depth mechanistic study to support authors main conclusion.
2. Among the patients studied in this manuscript, females had much higher risk of NEN than males before 55 years of age when sex hormones most profoundly affect gender differences in humans. There is no explanation for the role of sex hormones in this.
3. No explanation on why the gender impact on NEN survival are cancer site specific for NENs and whether the gender effects in the overall survival of the NEN patients are age-related.
4. No analysis on other factors, such as occupation, lifestyles, NEN treatment histories, and coexisting pathological conditions that could confound the results of gender impact on the overlay survival of male and female patients with NENs in the lung, rectum, pancreas, and stomach.
5. The table 3 is not clear. Kaplan Meier survival curves may be clear than this table.
6. Most female patients with NENs in the lung, rectum, pancreas, or stomach were diagnosed at earlier stage than male patients (Appendix). So, the treatment efficiencies rather than gender differences could impact the overall survival of patients with NENs in the lung, rectum, pancreas, or stomach. This need to be clarified. The survival of patient needs to be compared among patients diagnosed in the same stage.
Author Response
Please see attached response to reviewer 2.
Many thanks

Round 2
Reviewer 2 Report
There is no change in the revised version of the manuscript compared to the previous one. The authors only argue that this retrospective and population-based study that cannot address any questions raised from the last review.
In this case, the authors need to include all questions raised in the last review as well as the authors response to these questions in the discussion to explain 1) what the novelties of this study are, 2) what questions can or cannot be clearly answered by this study, 3) what are the possible pitfalls of using this database, and 4) what need to be done in epidemiological studies to properly address the role of sex hormone signaling pathways and NEN initiation, progression, and patients’ survival.
In addition, the results presented in this manuscript actually show that females have increased NEN tumorigenesis but better survival while males have lower risk of neuroendocrine neoplasia (NEN) but decreased survival than females after they are diagnosed with NEN. The authors should discuss from epidemiological point of view on what this means to studying the pathogenesis of NEN and how this could be validated in the future.
The authors claimed in the discussion that “We have demonstrated that females diagnosed with NEN tend to survive longer than males”. This sentence needs to be deleted as the authors did not “demonstrate” anything but only built this a link between gender difference and NEN survival based on a database of deceased patients that 1) lacks essential information needed to complete the proposed study and 2) could not be validated independent of cohorts. Therefore, any conclusion made in this manuscript needs to be confirmed and validated in the future prospective, randomized, and controlled clinical trials.